# Mechanisms of Aging and the Preventive Effects of Resveratrol on Age-Related Diseases

**DOI:** 10.3390/molecules25204649

**Published:** 2020-10-12

**Authors:** In Soo Pyo, Suyeon Yun, Ye Eun Yoon, Jung-Won Choi, Sung-Joon Lee

**Affiliations:** Department of Biotechnology, College of Life Sciences & Biotechnology, BK21-PLUS, Korea University, Seoul 02846, Korea; pyois95@korea.ac.kr (I.S.P.); suyun609@korea.ac.kr (S.Y.); yeeun916@korea.ac.kr (Y.E.Y.); cjay1@korea.ac.kr (J.-W.C.)

**Keywords:** aging, age-related diseases, resveratrol

## Abstract

Aging gradually decreases cellular biological functions and increases the risk of age-related diseases. Cancer, type 2 diabetes mellitus, cardiovascular disease, and neurological disorders are commonly classified as age-related diseases that can affect the lifespan and health of individuals. Aging is a complicated and sophisticated biological process involving damage to biochemical macromolecules including DNA, proteins, and cellular organelles such as mitochondria. Aging causes multiple alterations in biological processes including energy metabolism and nutrient sensing, thus reducing cell proliferation and causing cellular senescence. Among the polyphenolic phytochemicals, resveratrol is believed to reduce the negative effects of the aging process through its multiple biological activities. Resveratrol increases the lifespan of several model organisms by regulating oxidative stress, energy metabolism, nutrient sensing, and epigenetics, primarily by activating sirtuin 1. This review summarizes the most important biological mechanisms of aging, and the ability of resveratrol to prevent age-related diseases.

## 1. Introduction

Aging is a biological process defined as a decline over time in cellular and organismal functions, which also leads to various metabolic disorders including type 2 diabetes mellitus (T2DM) and cancer, and neurodegenerative diseases such as Parkinson’s disease (PD) and Alzheimer’s disease (AD) [1]. The hallmarks of aging are genomic instability, telomere attrition, epigenetic changes, loss of proteostasis, deregulated nutrient sensing, mitochondrial dysfunction, cellular senescence, stem cell exhaustion, and altered intercellular communication [2]. These are regarded as general factors and phenotypes of aging in respect to that they occur during normal aging, and experimental deterioration of them accelerates aging, while attenuation delays aging [2]. These processes are not necessarily independent of each other, and often occur simultaneously in an interconnected manner. Overall, however, the biology of aging is poorly understood [2]. Several approaches for improving healthspan have been suggested [3]. In this review, we examined several mechanisms that may be involved in the protective effects of resveratrol against age-related diseases.

## 2. Biological Mechanism of Aging

Aging is caused by endogenous and exogenous cellular stressors, which reduce the innate ability of cellular recovery. Aging is caused by changes in genetic information, chromosome structure, and protein homeostasis. For example, genomic damage, epigenetic alterations, telomere shortening, and impaired proteostasis are increased during aging of cells, tissues, and organisms, and damage accumulation in the cell can further accelerate the aging process. Cellular senescence, defined as irreversible cell cycle arrest, is another important characteristic of aging cells. In aging phenotypes, reactive oxygen species (ROS) show complex biphasic effects; although they may be beneficial at optimal levels, they can exert negative effects when present in excess [2]. Cellular senescence has some physiological benefits, such as repressing tumorigenesis; however, excessive accumulation of senescent cells exacerbates the negative effects of aging [4]. Due to the complex interconnection among aging phonotypes and processes, developing optimum anti-aging strategies has proven difficult. Recent results from phytochemical studies suggest a potential approach for slowing down the aging process in humans to reduce the risk of age-related diseases, which may improve lifespan and healthspan.

### 2.1. DNA Damage, Mutations, and Epigenetic Alterations during Aging

DNA damage and mutations are key features of aging cells both in vitro and in vivo. For example, it is well-known that Hutchinson–Gilford progeria syndrome is caused by a mutation in lamin A/C (*LMNA*) (1824C > T), which activates a cryptic splice site resulting in the expression of lamin with 50 amino acids deleted near its C-terminus. This deletion causes permanent farnesylation of the lamin protein, which deforms the nuclear membrane, causing loss of heterochromatin and changes to histone methylation [5]. Progeria syndrome, a genetic disease associated with a point mutation in the *LMNA* gene, shows how a single gene mutation can cause aging in humans. Mitochondrial DNA mutations also cause degenerative diseases and aging [6]. Mitochondrial DNA rearrangement mutations have been found in patients suffering from age-related diseases, including chronic coronary artery disease [7] and AD [8]. Thus, mutation and damage of both chromosomal and mitochondrial DNA can cause aging.

In cultured cells, DNA damage accumulates over time, further accelerating the aging process. Cells and organisms have endogenous DNA repair and ROS removal systems; however, these systems do not function effectively in aging cells. Thus, multiple damage to DNA or the repair system leads to an accumulation of DNA damage, resulting in epigenetic alterations that cause premature aging [9].

The telomere is a terminal region of repetitive DNA sequences at each end of the chromosome, which protects it from deterioration and prevents fusion with neighboring chromosomes. Telomere loss is a major biomarker of cell aging and telomere shortening can exacerbate age-related diseases [10]. A telomerase is required to maintain telomere length during DNA replication. A lack of telomerase function leads to loss of the nucleotide sequences that protect telomeres, and telomere shortening often results in reduced cell proliferation and cellular senescence [11]. Recent studies have revealed that short telomeres mediate hematopoietic and immune defects in patients with dyskeratosis congenita [12]. Short telomeres have a large effect on the replication capacity of stem cells in high-turnover tissue, such as bone marrow [13,14,15], thus contributing to aging acceleration. Experimental activation of telomerase delays or reverses aging phenotypes in mice [16]. These findings demonstrate that telomere is critical for protecting cells against the aging process.

In addition to mutations and damage to DNA, epigenetic alterations such as histone modification, chromatin remodeling, and DNA methylation occur progressively in the cells of aging individuals [17] and are associated with aging phenotypes and the development of age-related diseases. Since epigenetic alterations can be reversed, while genomic mutations cannot, reversing epigenetic alterations is a potential target for treatment aimed at delaying aging in cells [18]. In this context, sirtuin (SIRT) enzymes, which have NAD^+^-dependent protein deacetylases and show ADP ribosyltransferase activities, have been studied as a critical regulator of aging. For example, the upregulation of SIRT1 homologs such as *Saccharomyces cerevisiae* silent information regulator 2.1 (*Sir2.1*) in *Caenorhabditis elegans* [17], *dSir2* in flies [19], and SIRT6 in mice improves lifespan [20]. In addition, histone deacetylase inhibitors that reverse histone H4K12 acetylation attenuate age-related diseases and memory defects in aging mice [21]. These results suggest that either or both DNA damage and epigenetic changes to key genes can accelerate aging.

### 2.2. Accumulation of Damaged and Dysfunctional Protein with Aging

Metabolic and environmental stressors such as heat shock, endoplasmic reticulum (ER) stress, and oxidative stress cause aging due to the accumulation of damaged and dysfunctional proteins, in turn caused by defects in the autophagic pathway [22], proteasomal degradation [23], or chaperone-mediated folding [24]. During aging, proteins are progressively damaged and the capacity for proteostasis declines [25]. Impaired proteostasis networks, reflected in disrupted protein synthesis, folding, transport, condensation, and degradation, trigger age-related diseases such as AD and PD [26].

Heat shock proteins (HSPs) are protein chaperones that modulate protein folding in response to cellular stress and are also involved in aging [27]. Heat shock factor-1 (HSF-1) and DAF-16 transcription factors regulate HSPs. Moreover, their upregulation induces HSPs and extends the lifespan of *C. elegans* [28]. HSF-1 modulates HSPs via SIRT1 activity. SIRT1 deacetylates HSF-1, and deacetylated HSF-1 shows extended binding to the Hsp70 promoter to increase Hsp70 gene expression [29]. These findings suggest that transcriptional activation of HSPs can extend lifespan by inhibiting the aging process.

The activities of protein degradation systems, such as autophagy-lysosomal and ubiquitin-proteasome systems, decline progressively with aging [30,31]. The activation of autophagy can significantly extend lifespan in experimental animal models. For example, administration of rapamycin, a mammalian target of rapamycin complex 1 (mTORC1) inhibitor, induces macroautophagy and attenuates aging in mice [32]. Similarly, activation of the epidermal growth factor (EGF) signaling pathway and upregulation of proteins associated with the ubiquitin-proteasome system delay aging in *C. elegans* [33]. These findings suggest that activation of protein catabolism delays the aging process and may increase lifespan.

### 2.3. Energy Metabolism, Oxidative Stress, and Mitochondrial Functions in Aging

Caloric restriction alone extends the lifespan in mice, dogs, fish, invertebrate animals, and yeast [34]. Insulin and the insulin-like growth factor-1 (IGF-1) signaling (IIS) pathway, which is one of the nutrient sensing systems, are potential targets for delaying aging. Attenuation of the IIS pathway prolongs the lifespan by reducing cell growth, reproduction, and the metabolic rate, thereby storing energy for system maintenance [35].

Cellular levels of NAD^+^ and activation of NAD^+^ sensors play critical roles in the aging process. SIRT are NAD^+^-dependent deacetylases (class III histone deacetylases) that are highly conserved in various organisms. The first SIRT was discovered in yeast and named Sir2 [36]. In lower organisms, such as yeast, worms, and flies, SIRT-encoding genes are associated with lifespan extension [17,19,37]. Among the seven mammalian SIRTs (SIRT1–7), SIRT1 is the closest homolog to Sir2 based on its amino acid identity, and its activation leads to lifespan extension. Due to the promising results obtained in lower organisms, extensive research has been conducted to investigate the function of SIRT proteins in mammalian systems.

Nutrient sensing systems including AMP-activated kinase (AMPK), SIRTs, and mTOR also regulate aging [38]. AMPK is a master activator of catabolic pathways such as fatty acid β-oxidation, and suppresses anabolic metabolism, including cholesterol and fatty acid biosynthesis, thus improving insulin activity. Its activity is correlated with phosphorylation at threonine 172 (*p*-AMPK^Thr172^). Activation of AMPK has been proposed as a strategy for promoting longevity in mammals [39], and AMPK activity can be regulated in association with, or independent of SIRT activity. The ectopic expression of AMPK extends the lifespan in *C. elegans* [40]. Studies have suggested that decreasing anabolic signaling (IIS and mTOR) and increasing catabolic signaling (AMPK and SIRTs) via caloric restriction or caloric restriction mimetics can improve longevity [35].

Mitochondria generate oxygen radicals in the respiratory chain, and the efficiency of the electron transport chain progressively declines over time in oxygen-utilizing organisms [41]. According to the free radical theory, dysfunctional mitochondria lead to excessive ROS generation, thus inducing cell damage and aging [42]. Although oxidative stress is involved in diverse biological processes, including differentiation, immunity, autophagy, and metabolic adaptation, excessive levels of ROS exacerbate the aging process [43]. Accumulated ROS from dysfunctional mitochondria can cause activation of the inflammasome, and chronic ROS induction gives rise to age-related diseases [44]. Mitochondrial defects can induce permeabilization of the mitochondrial membrane and promote cell death by releasing cytochrome c and increasing the expression of pro-apoptotic proteins [41]. Low rates of mitochondrial removal and biogenesis due to multiple factors, including the accumulation of mutations in mitochondrial DNA, oxidized mitochondrial proteins, instability of the respiratory chain, and an imbalance between the fission and fusion of mitochondria, have a cumulative effect on aging [45]. However, mild mitochondrial stress caused by dietary interventions, caloric restriction, or physical exercise extends longevity by increasing stress resistance, which is also called mitochondrial hormesis or mitohormesis [46]. These findings suggest that nutrient sensing and energy metabolism are important in aging. Activation of catabolism and suppression of anabolism can delay aging. Mitochondria play a critical role in aging, and excessive ROS levels also accelerate the aging process.

### 2.4. Cellular Senescence

Cellular senescence is defined as irreversible arrest of cell proliferation in response to exogenous or endogenous stimuli. Cellular senescence promotes the clearance of damaged cells through apoptosis and the immune system, thus protecting tissues from oncogenesis [47]. However, the increased rate of senescent cell formation and decreased rate of their removal in aged individuals lead to the accumulation of senescent cells [48], which induce deterioration of the tissue microenvironment by secreting senescence-associated secretory phenotype (SASP) factors including inflammatory cytokines, interleukins, and growth factors [49]. Chronic inflammation by SASP factors exacerbates aging by disrupting macrophage functions, immune responses, and cell-to-cell communication [48]. Some tumor suppressors induce cellular senescence by regulating several pathways, including the cell cycle, DNA damage signaling, the immune response, and energy metabolism [50]. For example, tumor suppressor p16^INK4a^ (or p19^ARF^) is a cyclin-dependent kinase inhibitor that slows down cell proliferation, causing G1 arrest in the cell cycle. Increased levels of p16^INK4a^ (or p19^ARF^) have been observed in aging cells and are major biomarkers of cellular senescence [51]. In addition, inhibitors of the CDK 4a (INK4a)/ARF locus, which encodes p16^INK4a^ and p19^ARF^, are induced in senescent cells and cause age-related diseases including cancer, T2DM, glaucoma, and atherosclerotic diseases [52]. On the other hand, optimal activation of senescence-inducing tumor suppressor pathways can increase longevity [53]. Activation of p53 and INK4a/ARF inhibits the proliferation of damaged cells, conferring protection against oncogenesis [54]. These findings suggest that cellular senescence is a major phenotype of aging with biphasic effects. Control of senescence can suppress the aging process and tumorigenesis.

## 3. Role of Resveratrol in Slowing the Progression of Aging

Several genetic and environmental factors have been proposed as predisposing factors for early aging. Natural compounds and phytochemicals derived from plant sources have been investigated extensively in terms of their anti-aging properties. Excessive oxidative stress causes damage to DNA [55] and mitochondria, as well as proteostasis, thus accelerating the aging process [42]. In such cases, it is crucial to improve the ability of cells to maintain the ROS–antioxidant balance. Many dietary phytochemicals have ROS scavenging properties. Natural compounds with a phenolic component, including quercetin, resveratrol, and cyanidin, are antioxidant in nature. Their antioxidant activity is attributed to the presence of free hydroxyl groups that can donate hydrogen atoms to protect cells against lipid peroxidation [56]. Several phytochemicals, including resveratrol, potentiate endogenous anti-oxidative enzymes such as superoxide dismutase and catalase [57]. Among anti-aging phytochemicals, resveratrol is perhaps one of the most extensively studied; thus, we summarize its anti-aging properties in this review.

### 3.1. Chemical Properties of Resveratrol

Resveratrol (3,5,4′-trihydroxy-*trans*-stilbene) is a natural polyphenol with a molecular weight of 228.2 g/mol, placing it within the small molecule category. Resveratrol was first isolated in 1940 from the roots of *Veratrum grandiflorum* O. Loes (named white hellebore) [58]. It is also present in various plants such as grapes, cocoa, strawberries, tomatoes, peanuts, hop, cranberries, and sugar cane [59,60,61]. Regarding its chemical properties, resveratrol has a planar stilbene structure that gives it hydrophobic characteristics (Figure 1). Thus, it has high-affinity interactions with the hydrophobic domains of target protein molecules. In addition, the three polar hydroxy groups participate in hydrogen bonding with the amino acid side chains of target proteins; thus, resveratrol is a small-molecule ligand or regulator of direct protein interactions. To date, approximately 20 proteins (e.g., SIRT1 and nuclear factor erythroid 2-related factor2, Nrf2) have been shown to interact with resveratrol [62,63].

Regarding safety, many single-dose studies have been conducted to ascertain the safe daily dose of resveratrol. In a rat study, the potential toxicity of resveratrol orally administered at various doses (0, 300, 1000, and 3000 mg/kg body weight/day) for four weeks was examined. No adverse effect was observed at low doses (0–300 mg/day), while high doses (>1000 mg/day) caused kidney damage and body weight loss [64]. Several human studies found that oral administration of resveratrol (<1 g/day) does have any adverse effects in a short period (<1 month). However, a few adverse effects, such as abdominal pain and diarrhea, appeared when >0.5 g resveratrol was administered for one month [65]. The rate of absorption of resveratrol is quite high, and approximately 75% of orally administered resveratrol is absorbed in the human body. However, due to rapid metabolism of sulfonides and glucuronides conjugates in the liver and intestine, the oral bioavailability of resveratrol is low and these conjugates are eliminated in urine [66,67].

### 3.2. Resveratrol Activates SIRTs and AMPK

In a high-throughput in vitro screening study, resveratrol emerged as the most potent inducer of deacetylase activity among various polyphenols [68]. Subsequent studies have verified that resveratrol can extend the lifespan of *S. cerevisiae*, *C. elegans*, *Drosophila melanogaster*, and mice only if the gene that encodes the Sir2 homolog is present [68,69]. In addition, a previous study showed that resveratrol administration caused a significant decrease in brain protein content in aged fish models, extending the longevity by up to 59% [70]; however, it was not clear if the effect was Sir2-dependent. In humans, due to limitations regarding dosages and ethical issues, the effect of resveratrol on lifespan extension has not been proven.

Both in association with and independent of SIRTs, resveratrol activates AMPK, a critical regulator of energy metabolism and the aging process. Resveratrol activates AMPK in cultured cells without directly interacting with the AMPK protein [71]. Animal studies, especially rodent models, have demonstrated that resveratrol is an effective treatment for obesity and T2DM, improving glucose metabolism and lipid profiles, weight loss, and metabolic efficiency [65,72,73]. For example, in mice, resveratrol causes phosphorylation and thus improves AMPK activity, while also decreasing the expression of the enzymes responsible for lipogenesis and improving glucose uptake via translocation of glucose transporter type 4, which in turn improves insulin sensitivity [74].

Resveratrol directly binds to and activates SIRTs, which are cellular NAD^+^ sensors. One of the most important functions of resveratrol is the dose-dependent and reciprocal regulation of SIRT1 and AMPK. With low levels of resveratrol, SIRT1 deacetylates and activates liver kinase B1 (LKB1), which is an upstream kinase of AMPK, thus increasing AMPK activity. On the other hand, at high levels, resveratrol increases the cellular AMP-to-ATP ratio possibly by inhibiting mitochondrial ATP production, and thus activating AMPK. AMPK stimulates energy catabolism and increases cellular NAD^+^ levels, which further increases SIRT1 activity. AMPK is a negative regulator of mTOR, thus inducing autophagy and mitochondrial biogenesis, while SIRT1 inhibits nuclear factor kappa-light chain enhancer of activated B cells (NF-κB), leading to anti-inflammation and anti-cancer activity. The regulatory activity of resveratrol with respect to SIRT1 and AMPK has anti-aging effects. In addition, resveratrol can activate AMPK via cAMP and calcium-dependent mechanisms, which has a positive impact on aging and cellular senescence. Resveratrol inhibits phosphodiesterase (PDE), causing intracellular cAMP and Ca^2+^ levels to rise, consequently activating the AMPK pathway [59]. The AMPK pathway not only upregulates autophagy but also increases NAD^+^ levels, thus also increasing SIRT1 activity [59]. In addition, cAMP upregulates Nrf2 expression and induces antioxidant gene expression.

Thus, resveratrol is a potent activator of SIRT1 and a positive regulator of AMPK, increasing the lifespan in model organisms; however, the effect of resveratrol on human lifespan has not been studied. It may be reasonable to use surrogate markers of longevity and age-related diseases to evaluate the anti-aging effects of resveratrol.

### 3.3. Effects of Resveratrol on Age-Related Cardiovascular and Neurodegenerative Diseases

The cardioprotective effects of red wine were first reported in 1992 [75]. Since then, numerous studies have been conducted to understand the utility of resveratrol in the treatment of various pathophysiological diseases. To date, more than 110 clinical trials have been carried out on resveratrol mainly focusing on its effects on cardiovascular functions among other biological processes [59]. Resveratrol can slow down the aging process and is beneficial in the treatment of several diseases such as obesity, T2DM, cancer, cardiovascular diseases, and neurodegenerative diseases; it also acts as an immune system regulator [59,76,77,78], although there have been conflicting results on its effects on energy metabolism in humans [79]. Resveratrol acts on a variety of pathways involved in nutrient sensing and energy metabolism, as well as epigenetic modulation including the insulin/IGF-1, AMPK, mTORC1, and SIRT pathways. Thus, the mechanism of action of resveratrol with respect to its anti-aging properties is complex [76].

As described in the previous section, resveratrol is a direct activator of SIRTs and indirectly activates AMPK. AMPK is a negative regulator of mTOR, thus increasing autophagy and mitochondrial biogenesis. Autophagy impairment increases the risk of developing age-related neurodegenerative diseases such as PD and AD [59]. Based on multiple rodent studies, the neuroprotective role of resveratrol in AD, Huntington’s disease, PD, and neurological injury has been well established [73,74,80,81,82,83,84,85,86,87,88,89,90]. The levels of β-amyloid plaque, a marker of age-related changes in the brain, and its accumulation which causes AD are reduced with oral administration of resveratrol in humans [91]. These effects of resveratrol are largely explained by SIRT1 and AMPK activation.

The models of age-related eye disease, age-related macular degeneration, and autophagy impairment were seen in association with accumulation of the autophagy receptor p62. Accumulation of p62 is associated with activity in the AMPK pathway. AMPK phosphorylates and causes oligomerization of p62 during autophagosome formation, and oligomerized p62 accumulates in cells. Resveratrol can promote activation of the AMPK pathway and autophagy [59], reducing the accumulation of p62 and cellular waste and hence protecting against age-related diseases such as macular degeneration. Moreover, resveratrol can activate the SIRT1/PPAR γ co-activator 1 α pathway, which improves mitochondrial function and proteostasis [92].

Several animal studies have demonstrated that resveratrol can prevent T2DM and improve glucose metabolism and insulin sensitivity. For example, oral administration of resveratrol to mice fed a high-calorie diet improved their levels of insulin, glucose, and IGF-1, and the insulin sensitivity index, compared to high-calorie diet controls [93]. These findings suggest that activation of SIRTs and AMPK by resveratrol could be useful to prevent age-related diseases such as cardiovascular disease, T2DM, and neurodegenerative diseases.

### 3.4. Modes of Action of Resveratrol’s Anti-Cancer Activity

Several review articles have highlighted resveratrol’s anti-cancer activity [73,94,95,96,97,98], and numerous studies have evaluated the effects of resveratrol in combination with chemotherapeutic agents [98]. Resveratrol displays diversified mechanisms of anti-cancer activity to block all three stages of cancer (initiation, promotion, and progression). Among resveratrol’s anti-cancer mechanisms are the induction of apoptosis to remove damaged cells, improved anti-oxidant and detoxifying capacity (phase II enzymes), induction of cell cycle arrest, disabling angiogenic switches, reducing the blood and nutrition supply to tumorigenic sites, reduced metastasis and invasion, and sensitizing tumor cells when used in combination with another anti-cancer agent during chemotherapy [94,95,96,97]. Resveratrol induces apoptosis in leukemia, colon, breast, and prostate cancer cell lines.

Resveratrol has multiple targets in cancer cells, such as cyclooxygenase (COX)/lipoxygenase (LOX), tyrosine kinase (p56lck), protein kinase C (PKC), protein kinase D (PKD), mitogen-activated protein kinase (MAPK), extracellular signal-regulated kinase (ERK)1/2, stress-activated kinases c- Jun *N*-terminal kinase 1/2, p38 MAPK [99], and adenylyl cyclase.

There is evidence that resveratrol induces apoptosis in various cancer cells, but the underlying mechanism varies considerably among different cancer cell types [100]. Although the exact mechanism of resveratrol-induced apoptosis is not well defined, studies suggest that resveratrol triggers several apoptotic pathways. For example, expression of the pro-apoptotic B-cell lymphoma 2 (Bcl-2) associated X (Bax) gene by p53 or other transcription factors is stimulated by resveratrol-induced apoptosis in mitochondria [101,102,103]. In addition, mitochondria-mediated apoptosis may involve the downregulation of anti-apoptotic protein Bcl-2 [104], or translocation of Bax from the cytosol to mitochondria [105]. Resveratrol-induced activation of p53 is mediated by ERKs and p38 in the mouse JB6 epidermal cell line [106]. Moreover, resveratrol renders hepatoma G2 (HepG2) hepatocarcinoma cells sensitive to apoptosis by increasing ERK activity and decreasing the expression and activity of Akt, cyclin D1, and p21 activated kinase 1 [107].

COX and LOX activities are closely associated with cancer development, for example, *Cox2* gene deletion reduced colorectal cancer in a mouse model [108]. COX2 is believed to play a vital role in oncogenesis, as the inhibition of PKC, which regulates its activity, could prevent the development of cancer [109,110]. Resveratrol inhibits COX [111] and LOX activity, which is partly responsible for its anti-cancer effects [112]. Resveratrol suppresses COX1 protein levels but decreases COX2 mRNA levels [109,113,114]. Microsomal COX activity is significantly reduced in the lungs and hepatic system through resveratrol administration in vivo [113,115]. Thus, resveratrol can prevent the induction of COX and enhance cancer treatment.

Resveratrol can inhibit both COX and LOX, which can in turn induce the synthesis of proinflammatory molecules that are critical for the initiation of tumorigenesis. Thus, the inhibitory effect on COX/LOX activity is responsible for its anti-cancer properties [116]. In addition, resveratrol can inhibit p56lck, PKC, and PKD [117]. Inhibition of PKC induces growth inhibition and activation of apoptosis in a variety of cancer cell models, such as gastric cancer [118] and prostate cancer [119].

Inflammaging, chronic activation of the immune system, is one of the major aging processes in the brain [120]. Inflammation in the brain can occur due to oxidative damage, an excessive immune response to pathogens, the accumulation of proinflammatory cytokines in senescent cells, and dysregulation of autophagy [120]. Resveratrol activates SIRT1, which modulates physiological and metabolic responses to stress signals. SIRT1 inhibits phosphorylation of the p65 subunit of NF-kB through deacetylation of the RelA/p65 subunit at lysine 310. This leads to reduced transcription of the proinflammatory gene and hence anti-aging effects [120]. Inhibition of neuroinflammation by resveratrol may suppress brain aging and reduces the risk of neurodegenerative diseases. Further, resveratrol administration through the parenteral route can inhibit tumor growth in rodents [77].

### 3.5. Resveratrol Clinical Trials

Clinical data on the anti-aging effects of resveratrol on age-related diseases have increased gradually. Epidemiologic studies suggest that dietary intake of resveratrol decreases the likelihood of age-related diseases, particularly AD [121]. Of the several clinical trials on the effects of resveratrol on AD, the results from two trials suggest that resveratrol affects several AD biomarkers. In one study, resveratrol (500–1000 mg/day) was orally administered to patients with mild to moderate AD. The brain volume decreased with resveratrol for unknown reasons, but cognitive deterioration was not observed. Both in the resveratrol and placebo groups, Aβ40 (β-amyloid) levels in the cerebrospinal fluid and plasma were decreased at 52 weeks [121]. Another study reported similar results regarding Aβ40 levels in the cerebrospinal fluid compared to the control, suggesting that resveratrol can decrease Aβ40 levels and ameliorate the progression of AD. In plasma, resveratrol increases matrix metalloproteinase-9 (MMP-9) and MMP-10 and reduces interleukin-12P40 (IL-12P40) and IL-12P70 levels, which are typically increased in AD patients. These results suggest that resveratrol may regulate neuroinflammation and reduce β -amyloid accumulation in AD patients, but further studies are required to conclusively determine the efficacy of resveratrol in AD [122,123].

Clinical studies on the role of resveratrol in cancer have shown that resveratrol has a variety of targets within the cell, and its efficacy differs depending on the type and stage of cancer, dosage, and treatment period [124]. A clinical study indicated that resveratrol was safe to use in breast cancer patients. The participants were administered resveratrol for 12 weeks; as the trial progressed, it was found that the amount of resveratrol in blood serum samples increased. In addition, the trial revealed that resveratrol affected the epigenetic expression of Ras association domain family 1 isoform A, which is a gene associated with breast cancer, and this effect was correlated with the circulating levels of resveratrol [125]. These results suggest that resveratrol may act as a chemopreventive agent in breast cancer by influencing the epigenetics of breast cancer-associated genes; however, this finding needs to be confirmed in additional clinical trials. Another study on prostate cancer pathogenesis showed that resveratrol could delay cancer recurrence. After primary treatment, approximately 33–50% of prostate cancer patients experienced biochemical recurrence of the disease. Rising levels of prostate-specific antigen (PSA) are the earliest indication of disease recurrence. MPX (pulverized muscadine grape skin, which contains resveratrol) delays recurrence by increasing the PSA doubling time by 5.3 months, although these results were not statistically significant [126]. In contrast, several clinical trials failed to clearly demonstrate the effects of resveratrol on various cancers. For example, in clinical trials of patients with colorectal cancer, the results seem promising, but remain inconclusive regarding whether resveratrol could be a viable treatment. After about one–two weeks of treatment with resveratrol, the concentrations of parent resveratrol and its major metabolites in the colorectal tissue of patients were similar to the effective concentrations of resveratrol used in preclinical studies [127,128]. Although in vitro studies have shown that resveratrol inhibits growth and induces apoptosis in human colon cells, the anticarcinogenic activity of resveratrol’s metabolites has yet to be experimentally verified [115].

Clinical studies on the effects of resveratrol on blood flow, T2DM, and metabolic syndromes have been conducted. In a randomized, double-blind, placebo-controlled, crossover trial, the combination of piperine and trans-resveratrol significantly improved central blood flow in healthy young adults [129]. Statistically significant improvements in performance on a single cognitive task and subjective ratings of fatigue were observed in healthy young adults who received trans-resveratrol and piperine for 28 days [129]. In another trial, a significant increase in diastolic blood pressure was observed in a resveratrol group [130]. Furthermore, several clinical studies have demonstrated the roles of resveratrol in improving insulin resistance and metabolic syndrome [131,132,133,134,135]. A meta-analysis of 11 randomized clinical trials on the effects of resveratrol on glucose metabolism and insulin sensitivity revealed a significant effect in diabetic, but not non-diabetic, participants. The resveratrol dose varied from 8 to 1500 mg/day and was maintained from two weeks to six months. Diabetic patients showed meaningful reductions in fasting glucose (−35.22 mg/dL; *p* < 0.01), fasting insulin (−4.55 microunits/mL; *p* < 0.01), hemoglobin A1c (−0.79%; *p* = 0.02), and insulin resistance (homeostatic model assessment—insulin resistance; −2.25; *p* < 0.01). Significant heterogeneity was observed in blood sugar levels of diabetic patients [136]. Similar results were reported by a small randomized, double-blind, placebo-controlled trial of adult patients with metabolic syndrome and a body mass index (BMI) between 30 and 39. A treatment group received a 1500 mg daily dose of resveratrol for 90 days. There was a significant decrease in total weight, BMI, fat mass, waist circumference, and insulin secretion in the treatment group compared to the placebo group [137].

## 4. Conclusions, Limitations, and Future Perspectives

Aging is an inevitable and natural biological process that can cause several age-related diseases such as cancers, T2DM, cardiovascular disease, and neurological diseases. Mitigating the aging process can promote human health and may increase lifespan and healthspan. Conventional dietary interventions have been developed to prevent age-related diseases and could also increase lifespan. It is only very recent to have explored the effects of phytochemicals on lifespan and healthspan by directly targeting the aging process. Therefore, most phytochemical studies performed to date have focused on the effects on molecular and physiological aspects of age-related diseases. Resveratrol is an exceptional phytochemical, in that it has been shown to increase lifespan in several model organisms and is a potent activator of SIRT1. Resveratrol can slow the aging process by nutrient sensing (NAD^+^ detection) and regulating mitochondrial function, cellular senescence, and cancer cell proliferation/apoptosis/inflammation (Figure 2). Many resveratrol studies also showed somewhat conventional results from phytochemical research by investigating molecular markers for age-related diseases such as antioxidative activity. Several human trials have suggested the potential health benefits of resveratrol per se or in combination with other compounds such as piperine. Although there are several promising in vitro results, the effects of resveratrol on cancer have not been conclusively shown in human trials. However, resveratrol may be effective for preventing, or even treating metabolic diseases such as T2DM. Resveratrol may also be useful in preventing cardiovascular and neurodegenerative diseases, although long-term trials are required to verify its efficacy and safety. Furthermore, clinical trials investigating the effects of resveratrol on the major surrogate markers of the human healthspan are needed.

Although resveratrol is one of the most promising natural compounds for the prevention and treatment of age-related diseases through potent biological activities, there are several barriers in its application, the most important being its low bioavailability and rapid hepatic metabolism. The oral bioavailability of resveratrol is less than 1%, because it is metabolized rapidly into sulfate and glucuronide metabolites in the intestine and liver [138]. Moreover, the aqueous solubility of resveratrol is less than 1 mg/mL, which is a disadvantage for drug encapsulation [139]. A novel drug delivery system will help to overcome such shortcomings. Additionally, resveratrol has multiple target proteins and its activity is somewhat non-specific to protein isoforms, thus it is listed on pan-assay interference compounds (PAINS) [140] as a pan-histone deacetylase inhibitor with moderate inhibitory effects on the expression of histone deacetylases 1–11 [141]. The histone deacetylase family of enzymes is considered a promising target for cancer treatment [142]. It is unclear whether mild inhibition by resveratrol at relatively high concentrations would have significant effects, but it remains an interesting direct binding partner. More studies are required to confirm the biological activities, and effective and acceptable doses of resveratrol for long-term intake. Many PAINS are generally safe despite their interaction with multiple protein targets since they are natural compounds [143]. Thus, resveratrol is not an approved drug and may not be approved in the future; however, it may be used as an important functional food and food supplement. Food items containing high levels of resveratrol may prevent age-related diseases among the general public.

## Figures and Tables

**Figure 1 molecules-25-04649-f001:**
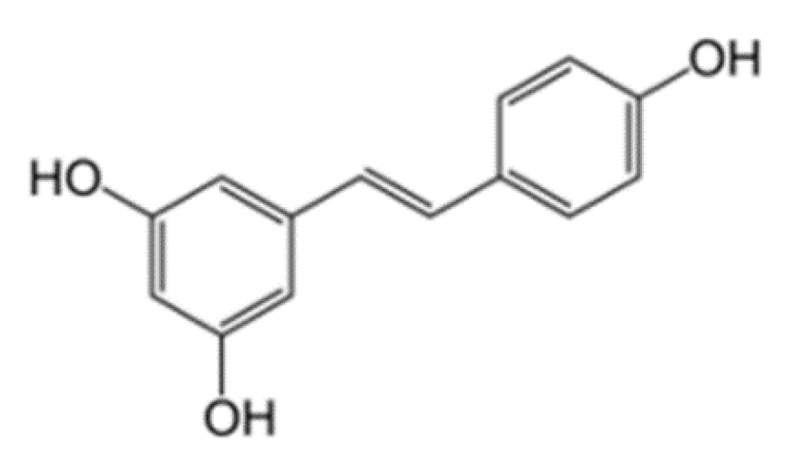
The structure of resveratrol.

**Figure 2 molecules-25-04649-f002:**
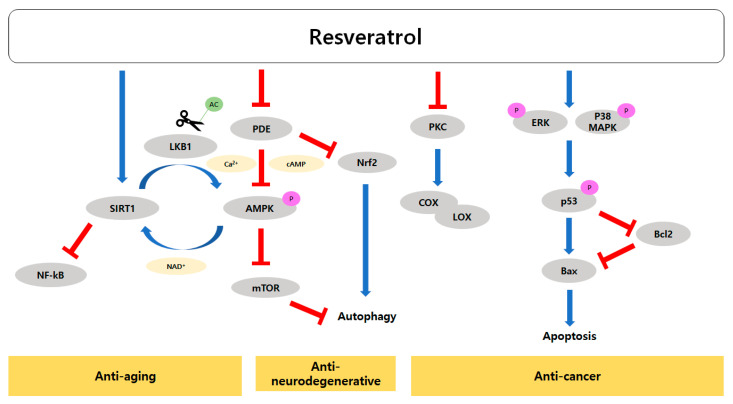
The anti-cancer, anti-neurodegenerative, and anti-aging effects of resveratrol are exerted through intracellular signal transduction pathways. Inhibition of PKC by resveratrol in turn induces the inhibition of COX and LOX, which can induce the synthesis of proinflammatory molecules that are critical for the initiation of tumorigenesis. Further, the expression of the pro-apoptotic Bax gene is stimulated by p53 or other transcription factors, and downregulation of the anti-apoptotic protein Bcl-2 leads to mitochondria-mediated apoptosis. Resveratrol can also inhibit PDE, which increases the concentrations of cAMP and Ca^2+^. The increase in cAMP and Ca^2+^ promotes AMPK and Nrf2, respectively, resulting in the activation of autophagy through the inhibition of mTOR. AMPK increases cellular NAD^+^ levels, which further promotes SIRT1 activity. Resveratrol can activate SIRT1, thus inhibiting phosphorylation of the p65 subunit of NF-kB. This causes a reduction in transcription of the proinflammatory gene and inhibition of ROS and cytokine production, leading to anti-aging effects.

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
