# Peer review of "Mechanisms of Aging and the Preventive Effects of Resveratrol on Age-Related Diseases"

_molecules, 2020, doi:10.3390/molecules25204649_

Round 1

Reviewer 1 Report

Dear Editor,

The current manuscript described a review on resveratrol with special focus on mechanism of aging and the preventive effects. Review is very interesting. The subject of considered manuscript fitting well with the scope of "Molecules" and indicated excellent scientific value.

My comments are as follows:

  1. Discuss the results of PAINS (Pan-Assay INterference compounds) or Badapple Promiscuity Analysis Output for resveratrol. There is a possibility (planar stilbene) that resveratrol can show various activities in in vitro assays due nonspecific binding with multiple biological targets.
  2. Include the negative and limitations of resveratrol in a separate section.
  3. Is resveratrol an approved drug by any regulatory agency if not also include it?
  4. Expert opinion is missing. Authors should include any successful approach (formulation or prodrug) which can be implemented for the successful development of resveratrol in clinical settings.

Regards,

A

Author Response

Reviewer #1

The current manuscript described a review on resveratrol with special focus on mechanism of aging and the preventive effects. Review is very interesting. The subject of considered manuscript fitting well with the scope of "Molecules" and indicated excellent scientific value.

Response: The manuscript has been intensively revised. All changes are highlighted in yellow.

My comments are as follows:

  • Discuss the results of PAINS (Pan-Assay INterference compounds) or Badapple Promiscuity Analysis Output for resveratrol. There is a possibility (planar stilbene) that resveratrol can show various activities in in vitroassays due nonspecific binding with multiple biological targets.

Response: Information of resveratrol about PAINS has been added (Lines 505-514)

  • Include the negative and limitations of resveratrol in a separate section.

Response: Limitation of resveratrol is discussed. (Lines 500-514)

  • Is resveratrol an approved drug by any regulatory agency if not also include it?

Response: Information is added (Lines 512)

  • Expert opinion is missing. Authors should include any successful approach (formulation or prodrug) which can be implemented for the successful development of resveratrol in clinical settings.

Response: Expert opinion has been added in section 4 (Line 483-514)

Reviewer 2 Report

Overall, the writing of the manuscript must be improved (in some parts of the text the inaccurate use of the language makes the text more difficult to read/follow) and in some cases, the authors are advised to re-write some sentences.

Moreover, in several parts of the article the information provided is confuse. The authors mix pretty often results obtained/observations made in worms and then speak about rodents... I believe that on each section the information could be better structured: results obtained in insects, rodents, humans... Indeed, in some parts of the manuscript I had to re-read the sentences in order to make sure whether the authors were referring to an experimental model or to data obtained in humans.

With regard to the usage of abbreviations, many have been directly included in the text, without writing the full names in their first appearance. The authors are encouraged to check and correct these aspects of the manuscript. Similarly, no numbering of the sections beyond “2. Mechanisms of action” is included. I believe that the section referring to the effects of resveratrol should be the 3rd, while the summary the 4th.

In addition, the authors should update the bibliographic references used in this manuscript (some references are referred to as “recent studies”, when they were published prior to 2010).

Abstract

Despite the word limitation, the authors should check some of the sentences included in this section. Indeed, too many short sentences are used, which could be put together in longer sentences with “more” content. Moreover, some inaccurate vocabulary has also been used (complicated, decelerate...), so the authors are encouraged to check/correct it. Finally, I would re-write the last sentence of the abstract in order to better match the title of the article (I would mention the word mechanisms instead of characteristics).

Introduction

Lines 32-34: Please check this sentence and re-write it (it is not clearly explained).

Lines 36-37: The information included in this sentence has already been given previously (line 28). The authors should choose where they want to include this information while avoiding repeating it.

Lines 38-41: The sentences included in this part of the text don´t make much sense. It is not clear why diet (only) is mentioned as an important factor related to aging process regulation. The authors should check this part and decide whether re-write or remove it.

Lines 42-43: Before start talking about the DNA damage etc theories of aging, a short introductory paragraph would allow the reader to better understand the current scenario (several theories have been proposed, each with strengths and weaknesses, to explain these complex and not fully understood process). The authors are encouraged to do so.

Mechanisms of aging

DNA damage and epigenetic alterations during aging

Extensive corrections are needed throughout this section of the manuscript. In some cases, the incorrect usage of the language make some sentences/concepts confuse and difficult to understand. In other parts, several short and unconnected sentences are put in a row. The authors must take into account the comments below in order to improve the quality/content of this section of the manuscript.

Lines 66-68: The authors give some kind of conclusion regarding what has been side till that point. I believe that when talking about a mechanism of action, these kind of statements are not appropriate. This sentences may better be fitted in the end of this section, as some kind of “final conclusion” regarding DNA damage relationship with the aging process (as it is done in the next section of the manuscript).

Line 70: I believe that “over a period of time” is not accurate enough. Please check the sentence and re-write it.

Lines 73-75: The sentence is not well written. The authors must check and correct it.

Damaged and dysfunctional protein accumulation on aging

Lines 88-91: Please re-write this sentence, it is too long/tricky to read/understand and the organization of the ideas is too messy.

Energy metabolism, oxidative stress, and mitochondrial functions on aging

Too many ideas are put together in this section of the manuscript, and again these ideas are not efficiently ordered. It seems that the authors want to explain lots of processes but then, as soon as an explanation is given, the start explaining other process. I believe that the information included in this section must be improved and better organized.

Lines 116-117: Same comment than that for lines 66-68.

Lines 118-119: This sentence does not make sense (ROS are produced in the mitochondria all the time, not only under caloric/energy restriction). The authors must check/re-write it.

Lines 119-120: This statement is inaccurate; ROS are constantly produced within the mitochondria. What happens is that the ROS production is greater in dysfunctional mitochondria. The authors must take this into account and check/correct the sentence.

Lines 130-133: The authors are proposing/formulating some sort of relationship among energy metabolism/mitochondrial (dys)function/aging, but using a too vague language. This part needs to be better explained or removed from the manuscript.

Lines 133-134: Including resveratrol at this point does not make sense. This section of the manuscript is devoted to explaining the mechanisms of action described to date to explain the aging process (or at least I believe that this was the original idea of the authors). Please remove this sentence from this part of the manuscript.

Cellular senescence

Same comment regarding the content/structure for this section as for the previous one.

Lines 141-143: Please check and re-write the sentence.

Resveratrol on aging prevention

After reading this part of the manuscript, I have the feeling that the authors didn´t have a clear idea about how they wanted to write this review. It seems that different/independent parts regarding resveratrol or its effects in various diseases have been written apart and then put together without noticing that the different sections within this section of the manuscript do not have an unifying thread.

The following is a proposal for the authors:

- Start with an introduction about resveratrol (when/who discovered the molecules, structure, chemical characteristics, food sources, bioavailability/limitations on its use...).

- Then talk about the different age-related diseases in which resveratrol has shown to exert beneficial effects (explaining the current knowledge and the involved mechanisms).

- Finally, talk about the clinical trials.

I do believe that in such way it would be easier for the reader to understand the information provided by the authors.

Lines 161-162: Please check and reformulate this sentence.

Lines 169-171: This sentence/information should be included in other part within this same section of the manuscript.

Line 174: Phenolic compound or polyphenol, but not polyphenol compound.

Line 178: The authors refer to figure 1 when talking about resveratrol structure. However, the image included in this figure is too small to clearly appreciate the chemical characteristics of the compound. The authors are encouraged to include a figure showing solely resveratrol´s chemical structure (figure 1), besides the figure that has been already included (then it would be figure 2).

Lines 184-188: The authors should better explain the information included in this part of the manuscript. In addition, the authors should specify where these findings have been made (using animals, humans...).

Resveratrol’s effects on age-related diseases

Lines 193-196: Please place all the references used in this sentence at the end of it.

Lines 200-211: The effects of resveratrol on neurodegenerative diseases and type 2 diabetes mellitus are mixed in the same paragraph. Regarding neurodegenerative diseases, no further information is given beyond stating that resveratrol has beneficial effects in rodents (what about humans?).

In the case of type 2 diabetes/insulin resistance, the authors mention a study, while in the literature are plenty of them showing the effects of resveratrol in glucose homeostasis (not only in rodents, but also in humans).

The authors should better organize this section.

Lines 224-262: These two paragraphs could/should be unified.

Lines 224-237: Again, I believe that such an interesting/complex topic is explained in a too simplistic way. It should be improved.

Lines 239-241: These two sentences are very similar and should be unified.

Nutrient sensing, energy metabolism, and epigenetic alterations by resveratrol

In the way it has been written, this part of the manuscript is confusing and the reader can get lost while reading it. The authors should better select/decide the information they want to provide and then organize/write it properly. I strongly recommend to re-write this whole section.

Lines 279-295: In the same paragraph the authors write about resveratrol and energy sensing, autophagy and the effects of resveratrol in eye disease. Honestly, I believe that the authors should better plan what they want to include on each section and then start writing it.

Resveratrol clinical trials

I have a similar feeling with this section. In the same paragraph the authors talk about healthy subjects treated with resveratrol to analyse whether this compound exerts beneficial effects on blood pressure, and about subjects with Alzheimer disease and cognitive outcomes of the treatment with the polyphenol. Again, I encourage the authors to re-write this part of the manuscript since on its current form is confusing and difficult to follow/understand.

Summary

Lines 362-363: This sentence is incomplete.

Author Response

Reviewer #2

Overall, the writing of the manuscript must be improved (in some parts of the text the inaccurate use of the language makes the text more difficult to read/follow) and in some cases, the authors are advised to re-write some sentences.

Moreover, in several parts of the article the information provided is confuse. The authors mix often results obtained/observations made in worms and then speak about rodents. I believe that on each section the information could be better structured: results obtained in insects, rodents, humans... Indeed, in some parts of the manuscript I had to re-read the sentences in order to make sure whether the authors were referring to an experimental model or to data obtained in humans.

With regard to the usage of abbreviations, many have been directly included in the text, without writing the full names in their first appearance. The authors are encouraged to check and correct these aspects of the manuscript. Similarly, no numbering of the sections beyond “2. Mechanisms of action” is included. I believe that the section referring to the effects of resveratrol should be the 3rd, while the summary the 4th. (numbered in each section)

In addition, the authors should update the bibliographic references used in this manuscript (some references are referred to as “recent studies”, when they were published prior to 2010).

Response: We intensively revised manuscript according to the reviewer’s comments. All changes are highlighted in yellow. We clarified several sentences whether the results were from insects, rodents, or humans. We changed numbering of each section as reviewer suggested. Some old references were not mentioned as “recent studies”. Thanks for the comments.

Abstract

Despite the word limitation, the authors should check some of the sentences included in this section. Indeed, too many short sentences are used, which could be put together in longer sentences with “more” content. Moreover, some inaccurate vocabulary has also been used (complicated, decelerate...), so the authors are encouraged to check/correct it. (Lines 13-15) Finally, I would re-write the last sentence of the abstract in order to better match the title of the article (I would mention the word mechanisms instead of characteristics). (Lines 21-23)

Response: We revised abstract intensively according to the reviewer’s comments.

Introduction and Mechanisms of aging

Response: This section has been intensively revised. Unclear sentences and expressions were changed. Additional explanation was added and redundant sentences were removed. 

Lines 32-34: Please check this sentence and re-write it (it is not clearly explained).

Response: Re-written (Lines 32-34)

Lines 36-37: The information included in this sentence has already been given previously (line 28).  The authors should choose where they want to include this information while avoiding repeating it.

Response: Sentence has been deleted.

Lines 38-41: The sentences included in this part of the text don´t make much sense. It is not clear why diet (only) is mentioned as an important factor related to aging process regulation. The authors should check this part and decide whether re-write or remove it.

Response: Changed (Lines 36-39)

Lines 42-43: Before start talking about the DNA damage etc theories of aging, a short introductory paragraph would allow the reader to better understand the current scenario (several theories have been proposed, each with strengths and weaknesses, to explain these complex and not fully understood process). The authors are encouraged to do so.

Response: Revised (Lines 42-53)

DNA damage and epigenetic alterations during aging

Extensive corrections are needed throughout this section of the manuscript. In some cases, the incorrect usage of the language makes some sentences/concepts confuse and difficult to understand. In other parts, several short and unconnected sentences are put in a row. The authors must consider the comments below in order to improve the quality/content of this section of the manuscript.

Response: Thanks for the comments. We intensively revised the text according to the reviewer’s comments.

Lines 66-68: The authors give some kind of conclusion regarding what has been side till that point. I believe that when talking about a mechanism of action, these kind of statements are not appropriate. These sentences may better be fitted in the end of this section, as some kind of “final conclusion” regarding DNA damage relationship with the aging process (as it is done in the next section of the manuscript).

Response: Deleted.

Line 70: I believe that “over a period of time” is not accurate enough. Please check the sentence and re-write it.

Response: Changed (Line 77)

Lines 73-75: The sentence is not well written. The authors must check and correct it.

Response: Revised (Lines 82-83)

Damaged and dysfunctional protein accumulation on aging

Lines 88-91: Please re-write this sentence, it is too long/tricky to read/understand and the organization of the ideas is too messy.

Reponse: Re-written (Lines 94-96)

Energy metabolism, oxidative stress, and mitochondrial functions on aging

Too many ideas are put together in this section of the manuscript, and again these ideas are not efficiently ordered. It seems that the authors want to explain lots of processes but then, as soon as an explanation is given, the start explaining other process. I believe that the information included in this section must be improved and better organized.

Lines 116-117: Same comment than that for lines 66-68.

Response: Deleted.

Lines 118-119: This sentence does not make sense (ROS are produced in the mitochondria all the time, not only under caloric/energy restriction). The authors must check/re-write it.

Response: Revised (Lines 121-122)

Lines 119-120: This statement is inaccurate; ROS are constantly produced within the mitochondria. What happens is that the ROS production is greater in dysfunctional mitochondria. The authors must take this into account and check/correct the sentence.

Response: Revised (Lines 122-123)

Lines 130-133: The authors are proposing/formulating some sort of relationship among energy metabolism/mitochondrial (dys)function/aging, but using a too vague language. This part needs to be better explained or removed from the manuscript.

Response: Deleted.

Lines 133-134: Including resveratrol at this point does not make sense. This section of the manuscript is devoted to explaining the mechanisms of action described to date to explain the aging process (or at least I believe that this was the original idea of the authors). Please remove this sentence from this part of the manuscript.

Response: Deleted.

Cellular senescence

Same comment regarding the content/structure for this section as for the previous one.

Lines 141-143: Please check and re-write the sentence.

Response: Revised (Lines 139-145)

Resveratrol on aging prevention

After reading this part of the manuscript, I have the feeling that the authors didn´t have a clear idea about how they wanted to write this review. It seems that different/independent parts regarding resveratrol or its effects in various diseases have been written apart and then put together without noticing that the different sections within this section of the manuscript do not have an unifying thread. (The resveratrol part(#3) was reorganized into major 5 sections. Chemical properties/effects on anti-aging/age-related diseases/anti-cancer/clinical trials)

The following is a proposal for the authors:

- Start with an introduction about resveratrol (when/who discovered the molecules, structure, chemical characteristics, food sources, bioavailability/limitations on its use...).

- Then talk about the different age-related diseases in which resveratrol has shown to exert beneficial effects (explaining the current knowledge and the involved mechanisms).

- Finally, talk about the clinical trials.

I do believe that in such way it would be easier for the reader to understand the information provided by the authors.

             Response: This section has been intensively revised. All changes are shown in yellow.

Lines 161-162: Please check and reformulate this sentence.

Response: Revised (Lines 158-159)

Lines 169-171: This sentence/information should be included in other part within this same section of the manuscript.

Response: Deleted

Line 174: Phenolic compound or polyphenol, but not polyphenol compound.

Response: Changed (Line 172, ‘compound’ deleted)

Line 178: The authors refer to figure 1 when talking about resveratrol structure. However, the image included in this figure is too small to clearly appreciate the chemical characteristics of the compound. The authors are encouraged to include a figure showing solely resveratrol´s chemical structure (figure 1), besides the figure that has been already included (then it would be figure 2).

Response: Figures have been revised. Figure 1 is a chemical structure and Figure 2 is a summary of resveratrol mode of action

Lines 184-188: The authors should better explain the information included in this part of the manuscript. In addition, the authors should specify where these findings have been made (using animals, humans...).

Response: Revised (Lines 182-192)

Resveratrol’s effects on age-related diseases

Lines 193-196: Please place all the references used in this sentence at the end of it.

Response: References were shifted into the end of this sentence

Lines 200-211: The effects of resveratrol on neurodegenerative diseases and type 2 diabetes mellitus are mixed in the same paragraph. Regarding neurodegenerative diseases, no further information is given beyond stating that resveratrol has beneficial effects in rodents (what about humans?).

Response: Information about human was added (Lines 247-251)

In the case of type 2 diabetes/insulin resistance, the authors mention a study, while in the literature are plenty of them showing the effects of resveratrol in glucose homeostasis (not only in rodents, but also in humans).

Response: Revised (Lines 260-269)

The authors should better organize this section.

Response: We intensively revised the manuscript re-organizing paragraphs.

Lines 224-262: These two paragraphs could/should be unified.

Response: Revised (Lines 282-329)

Lines 224-237: Again, I believe that such an interesting/complex topic is explained in a too simplistic way. It should be improved.

Response: Information was added (Lines 282-306)

Lines 239-241: These two sentences are very similar and should be unified.

Response: Revised (Lines 307-308)

Nutrient sensing, energy metabolism, and epigenetic alterations by resveratrol

In the way it has been written, this part of the manuscript is confusing and the reader can get lost while reading it. The authors should better select/decide the information they want to provide and then organize/write it properly. I strongly recommend to re-write this whole section.

Response: This section has been intensively revised.

Lines 279-295: In the same paragraph the authors write about resveratrol and energy sensing, autophagy and the effects of resveratrol in eye disease. Honestly, I believe that the authors should better plan what they want to include on each section and then start writing it.

Response: This section has been changed

Resveratrol clinical trials

I have a similar feeling with this section. In the same paragraph the authors talk about healthy subjects treated with resveratrol to analyse whether this compound exerts beneficial effects on blood pressure, and about subjects with Alzheimer disease and cognitive outcomes of the treatment with the polyphenol. Again, I encourage the authors to re-write this part of the manuscript since on its current form is confusing and difficult to follow/understand.

Response: This section has been intensively revised. 

Summary

Lines 362-363: This sentence is incomplete.

Response: Sorry for the errors. Errors have been corrected. (Lines 418-419)

Round 2

Reviewer 2 Report

The authors have taken into account the comments/suggestions made regarding the previous version of the manuscript, and therefore I do believe that the manuscript has been imrpoved. However there are still some sentences that could be better written (I guess that when the authors are requested to proof-read the manuscript could be fixed).

Author Response

We have proof-read our paper and made effort to improve the sentences. Thank you for your advice.